# Fabrication and Characterization of Sulfonated Graphene Oxide (SGO) Doped PVDF Nanocomposite Membranes with Improved Anti-Biofouling Performance

**DOI:** 10.3390/membranes11100749

**Published:** 2021-09-29

**Authors:** Muhammad Zahid, Tayyaba Khalid, Zulfiqar Ahmad Rehan, Talha Javed, Saba Akram, Anum Rashid, Syed Khalid Mustafa, Rubab Shabbir, Freddy Mora-Poblete, Muhammad Shoaib Asad, Rida Liaquat, Mohamed M. Hassan, Mohammed A. Amin, Hafiz Abdul Shakoor

**Affiliations:** 1Department of Chemistry, University of Agriculture, Faisalabad 38000, Pakistan; rmzahid@uaf.edu.pk (M.Z.); tayyabakhalid59@gmail.com (T.K.); 2Department of Materials, National Textile University, Faisalabad 37610, Pakistan; rubabagronmy123@yahoo.com; 3College of Agriculture, Fujian Agriculture and Forestry University, Fuzhou 350002, China; 4Department of Agronomy, University of Agriculture, Faisalabad 38000, Pakistan; shoaib.asad3@gmail.com; 5Department of Textile Engineering, National Textile University, Faisalabad 37610, Pakistan; saba.akram1980@gmail.com (S.A.); anumrashid800@yahoo.com (A.R.); 6Department of Chemistry, Faculty of Science, University of Tabuk, Tabuk 47311, Saudi Arabia; Khalid.mustafa938@gmail.com; 7Institute of Biological Sciences, Campus Talca, Universidad de Talca, Talca 3465548, Chile; morapoblete@gmail.com; 8Department of Chemistry, Government College University, Sahiwal Campus, Faisalabad 57000, Pakistan; liaquatali4224@gmail.com; 9Department of Biology, College of Science, Taif University, P.O. Box 11099, Taif 21944, Saudi Arabia; khyate_99@yahoo.com; 10Department of Chemistry, College of Science, Taif University, P.O. Box 11099, Taif 21944, Saudi Arabia; mohamed@tu.edu.sa; 11Department of Chemistry, University of Okara, Okara 56181, Pakistan; liaquat4224@gmail.com

**Keywords:** membranes, anti-biofouling, sulfonated graphene oxide, sustainability

## Abstract

Emergence of membrane technology for effective performance is qualified due to its low energy consumption, no use of chemicals, high removal capacity and easy accessibility of membrane material. The hydrophobic nature of polymeric membranes limits their applications due to biofouling (assemblage of microorganisms on surface of membrane). Polymeric nanocomposite membranes emerge to alleviate this issue. The current research work was concerned with the fabrication of sulfonated graphene oxide doped polyvinylidene fluoride (PVDF) membrane and investigation of its anti-biofouling and anti-bacterial behavior. The membrane was fabricated through phase inversion method, and its structure and morphology were characterized by Fourier transform infrared spectroscopy (FTIR), scanning electron microscopy (SEM), X-rays diffraction (XRD) and thermo gravimetric analysis (TGA) techniques. Performance of the membrane was evaluated via pure water flux; anti-biofouling behavior was determined through Bovine Serum albumin (BSA) rejection. Our results revealed that the highest water flux was shown by M7 membrane about 308.7 Lm^−2^h^−1^/bar having (0.5%) concentration of SGO with improved BSA rejection. Furthermore, these fabricated membranes showed high antibacterial activity, more hydrophilicity and mechanical strength as compared to pristine PVDF membranes. It was concluded that SGO addition within PVDF polymer matrix enhanced the properties and performance of membranes. Therefore, SGO was found to be a promising material for the fabrication of nanocomposite membranes.

## 1. Introduction

Water is the primary need for the existence of life on earth. Rapid developments in industries have underprivileged the basic human right of accessing fresh water by polluting underground and surface water assets. The condition has become worse due to the rise in poverty, population growth and expensive water treatment technologies [1]. According to a rough estimation, about 633 million natives are forced to consume polluted water from surface reserves, leading them to have waterborne diseases due to non-availability of fresh water [2]. There is an urgent need to develop effective technologies for water treatment to eradicate the major water crises. Therefore, desalination of sea water and wastewater treatment technologies have been developed to produce fresh water to fulfill human needs [3]. For the last few decades, membrane-based separation techniques have been proven to be the most effective process for pre-treatment of sea water due to its low energy requirements, ease of operation, high removal capacity, cost effectiveness, high productivity, no phase changing and easy scaling-up. These features make the membrane technology more demanding for desalination and purification of protein solutions [4]. From water, the separation of organic materials, bacteria, suspended particles, colloidal particles and proteins is effectively performed through ultrafiltration (UF) membranes [5]. The membrane’s performance depends on the membrane material, permeability, hydrophilicity, surface charge, thickness and pore size including operation conditions, such as filtration time, pressure and feed solution concentration [6]. Polymer material is considered an excellent material for membrane fabrication due to extraordinary properties like high chemical tolerance, mechanical strength and thermal stability. Polyvinylidene fluoride (PVDF) is extensively used among all the polymers for the fabrication of ultrafiltration membranes due to properties like elevated thermal and mechanical stability, excellent membrane-forming properties and good chemical resistance [7]. PVDF exhibits good processability for the formation of flat sheet membranes due to its promising ability to be soluble in a variety of solvents like dimethylformamide (DMF), N,N-dimethylacetamide (DMAc) and N-methyl-2-pyrrolidone (NMP) [8]. Although PVDF is hydrophobic in nature, it makes the membrane more susceptible to biofouling. Biofouling is the cake layer formation due to accumulation of proteins, bacteria and other organic materials on the surface of the membrane [9] which decreases the water flux of the membrane [10]. 

To control membrane biofouling, membrane surface hydrophilicity is improved using advanced methods [11]. The membrane surface is modified by the addition of hydrophilic nanomaterials to improve hydrophilicity of PVDF membrane. This technique has gained more interest due to easy operations and usual operation conditions [12]. Different types of nanomaterials like TiO_2_, SiO_2_, Ag, Co, CuO, Fe_2_O_3_, ZrO_2_, MgO, Al_2_O_3_ and other carbon-based materials such as graphene oxide, carbon nanotubes and functionalized graphene have been applied for improving the water permeability, stability, antibacterial and anti-biofouling behavior of membranes [13]. In addition, conductive polymer can also be used as membrane material. However, as a filler nanomaterial, GO has given more importance in recent decades owing to its superiority such as greater surface area, stronger mechanical stability and chemical inertness. Oxygen holding functional moieties present on GO surface are employed for membrane surface modifications leading to charge-based separations [14]. However, graphene oxide has hydrophilic nature making the dispersion of graphene oxide easier in polymeric matrix. [15]. Functionalization of graphene oxide with sulfonic acid functional groups improves the hydrophilicity, dispersibility and antifouling behavior of membrane [16,17]. Sulfonic acid functional group imparts superior characteristics to graphene oxide, which improves the performance of PVDF-based ultrafiltration membranes [5]. Together with electronic insulation capability and elevated mechanical strength, sulfonation makes graphene oxide a robust multifunctional material. Sulfonated graphene oxide was synthesized through diazonium salt [18].

In the present research work, graphene oxide was first synthesized and functionalized with sulphonic group and then PVDF-based ultrafiltration membranes were fabricated by incorporating the different concentrations of sulfonated graphene oxide through phase inversion process. The membrane characterization and performance were analyzed to investigate the effect of sulfonated graphene oxide on the final application of membrane.

## 2. Materials and Methods

Polyvinylidene Fluoride (PVDF, MW, 180,000 Da) and Potassium Permanganate (KMnO_4_, MW, 158.03 g/mol) were purchased from Sigma Aldrich (St. Louis, MO, USA). Polyvinylpyrrolidone (PVP, MW 10K, 40K), Sulfanilic acid (C_6_H_7_NO_3_S, Mm, 173.19 g/mol) and N, N-dimethyl acetamide (DMAc, Mm, 87.12 g/mol) were purchased from DAEJUNG Chemicals (Siheung-si, Korea). Graphite powder (99.99%), H_2_SO_4_ (98.5%) and hydrogen peroxide (H_2_O_2_, 35%) were purchased from Sigma Aldrich (St. Louis, MO, USA).

### 2.1. Synthesis of Graphene Oxide (GO) and Sulfonated Graphene Oxide (SGO)

The preparation of graphene oxide was done through modified hummer’s method. The oxidation of graphite powder (5 g) was done by mixing it with sodium nitrate (2.5 g) and sulfuric acid (200 mL) under continuous stirring. Then potassium permanganate (30 g) was slowly added to the mixture under ice bath condition over a 5–6 h period. Then, the mixture was kept under continuous stirring for 5 days at 37 °C until the appearance of a dark brown color for the mixture. Afterwards, 200 mL of deionized water is added into the mixture by elevating its temperature up to 98 °C. Then, 30 mL H_2_O_2_ was added for the reduction of untreated permanganate which would change the color to bright yellow. The mixture was then kept for settling and centrifuged, followed by its drying in an oven at 60 °C [19].

The sulfonation of graphene oxide was done by diazonium salt of sulfanilic acid. Firstly, 100 mg of GO was added into 16 mL of sulfanilic acid (0.06 M) solution at about 70 °C. Under continuous shaking, 4 mL of 6 × 10^−3^ M sodium nitrate solution was then dropwise added into the flask and kept at 70 °C for 12 h. After that, the mixture was washed with water using a centrifuge machine until its neutral pH was attained [20].

### 2.2. Fabrication of Nanocomposite Membranes

For the fabrication of nanocomposite membranes, a phase inversion method was employed for which casting solutions were prepared [21]. First of all, different concentrations of SGO nanomaterials were dispersed in N-N, dimethyl acetamide (DMAc) solvent by sonicating the solution for about 2 h. Then, polyvinylidene fluoride (PVDF) polymer (15%) and polyvinylpyrrolidone (PVP) polymer (1%) which act as pore forming agent were added in solution through magnetic stirring for at least 2 h. The concentrations of SGO in casting solutions were kept at: 0, 0, 0.1, 0.2, 0.3, 0.4 and 0.5 wt% and these membranes were named as M1, M2, M3, M4, M5, M6 and M7, respectively, as shown in Table 1. After the complete mixing of polymer into solvent, the solution was stored at room temperature overnight to remove air bubbles. Membranes were then cast on a steel plate using membrane casting machine (POROMETER, model: MEMCASTPLUS^TM^, Berlin, Germany). After waiting for 30 s, a steel plate was immersed in a coagulation bath containing deionized water. After 2–3 min, the membrane solidified and peeled off the plate. The membrane thickness was kept around 250 μm. The prepared membrane after drying was stored at room temperature [22].

### 2.3. Characterization

The sulfonated graphene oxide was analyzed by FTIR spectrometer of PerkinElmer (Greenville, SC, USA) (spectra 100) whereas the crystal structure of the nanocomposite membranes was characterized by PANalytical X’pert X-ray diffraction (XRD) (Malvern, Worcestershire, UK). FTIR spectrometer was used for the confirmation of functional groups present on membrane surface. The spectra were recorded in the range of 400–4000 cm^−1^ using 32 scans and a resolution of 4 cm^−1^.

Scanning electron microscopy (FEI, Quanta FEG 450, Quorum Q150R ES, Quorum technologies Ltd. Lewes, UK) was used to characterize the morphological features of fabricated membranes. Thermogravimetric analysis (TGA) was conducted using TGA (NETZSCH-Gerätebau GmbH, Selb, Germany) to measure thermal stability of membranes. Thermograms were traced (heat ranging 25–700 °C at the rate of 10 °C/min) to estimate the thermodynamic nature of membranes. Mechanical properties of membranes were measured through universal testing unit (Model 3385H, Instron, Norwood, MA, USA) as per ASTM-D 882-01 with a crosshead speed of 10 mm min^−1^ [3].

### 2.4. Water Contact Angle and Porosity Measurement

The hydrophilicity was investigated by measuring contact angle (Attension Theta Tensiometer, Västra Frölunda, Sweden) through the sessile drop method. The porosity of membranes was measured by gravimetric method Equation (1):(1)$$\varepsilon \;\left(\%\right)=\frac{\left[\left(w1-w2\right)/Dk\right]}{\left[\frac{w1-w2}{Dk}\right]+\left[w_{2}/D_{Pol}\right]}$$
while *w*1 = wet membrane weight, *w*2 = dry membrane weight, *Dk* = density of kerosene oil (0.82 g/cm^3^), *Dpol* = Density of PVDF = 1.78 g/cm^3^ [23].

### 2.5. Pure Water Flux and Salt Rejection

Water permeability was conducted using filtration assembly at 1 bar operating pressure at room temperature. The water flux was measured by the following Equation (2):(2)$$J=\frac{V}{S\cdot t}$$
where *V* is the volume of permeate solution taken in the permeability test at the time *t*, *S* is area of the membrane and *t* is time employed during permeation. 

The tests for solute retention were conducted using a sea water solution. After permeation, the difference in volume or concentration of salts in feed solution and permeate solution was estimated by calculating conductivity of solutions. The rejection of salts was determined by the following Equation (3):(3)$$R\left(\%\right)=\left[1-\frac{\;\;Cp\;\;\;\;}{Cf}\right]\times 100$$
where *Cp* and *Cf* are the concentrations of salts in permeate and feed solutions, respectively [24].

### 2.6. Anti-Biofouling Test

For conducting anti-biofouling test of membranes, a 1000 ppm BSA solution was taken as model biofoulant. The permeation was done at 1 bar pressure. Before conducting the experiment, the ultrafiltration membrane was packed down at 2 bar pressure for 20 min to reach the steady flux, designated as *J_o_*. Consequently, the bovine serum albumin solution was filtered for twenty minutes, and the membrane was rinsed using distilled water to eliminate the deposited foulants on the surface of the membrane. At last, filtration of pure distilled water was carried out again for 20 min, designated as *J_R_*. The BSA absorbance was calculated at 280 nm by employing UV-spectrophotometer (Cecil CE 7200), then the water flux recovery ratio was designed by using the equation given below [25].
(4)$$F_{R}=\frac{J_{R}}{J_{0}}\times 100\%$$

### 2.7. Antibacterial Test

The antibacterial activity of membrane samples was investigated by disk diffusion method using Gram-positive bacteria, *Staphylococcu aureus* and Gram-negative bacteria, *Escherichia coli*. In this method, all bacterial culture was first refreshed by growing them on a nutrient medium. For this, nutrient broth was taken in a flask and 100 mL water was added to it. Then, the flask was held in the autoclave at 120 °C for 20 min. After that, media was kept for cooling at room temperature and bacterial culture was inoculated in flask. Then, the flask was kept in an orbital shaker for 24 h. The next day, the bacterial inoculum was grown or dispersed on a sterile petri dish. The sample of membranes were cut down into circular shapes and kept in petri dish having bacterial culture. Ampicillin was employed as a control sample. After incubating at 37 °C for 24 h under an aerobic environment, confluent bacterial growth in the vicinity of the membrane was observed to determine the anti-bacterial activity of membranes [26].

## 3. Results and Discussion

### 3.1. Characterization of Nanocomposite Membranes

The Fourier transform infrared spectrum was engaged to prove the well graphite oxidation into graphene oxide and further sulfonization of graphene oxide (GO) into sulfonated graphene oxide (SGO). There are various functional entities—for example, hydroxy, epoxy and carboxyl—that exist on the surface of graphene oxide.

Figure 1 demonstrates the stretching vibrations of C=C at 1618 cm^−1^ and the deformation of C–O peak at 1390 cm^−1^. The peaks present at 1727 cm^−1^, 1070 cm^−1^ and 1216 cm^−1^ proved the existence of carbonyl, alkoxy and epoxy functional moieties. All of these peaks confirmed the good graphite oxidation into graphene oxide (GO) [20]. Subsequent to reduction and sulfonization, the peaks present at 1060 cm^−1^, 1250 cm^−1^, and 1365 cm^−1^ are critically attenuated in the sulfonated graphene oxide (SGO) plot. The two peaks present at 1175 cm^−1^ and 1126 cm^−1^ (S-O) and peak at 1040 cm^−1^ (S-phenyl) confirms the existence of –SO_3_ sulfonic acid group, and peaks present at 1007 cm^−1^ for C–H in-plane deformation and at 833 cm^−1^ out-of-plane H wagging are distinguishing vibrational peaks of a p-di substituted phenyl group as represented in Figure 2 [27]. 

XRD method is used to characterize the microstructure of nanomaterials, i.e., graphene oxide and sulfonated graphene oxide. On oxidation of graphite into graphene oxide, the diffraction peak shifted to 2θ = 10.6° from 2θ = 26°, representing the expansion of interlayer spaces by inclusion of oxygen holding functional groups, i.e., carboxyl, epoxy and carbonyl as described in Figure 3. Together, the weak peaks present at 2θ = 55° vanished. There is no major variation observed among the XRD plot of graphene oxide and sulfonated graphene oxide, showing that the presence of sulfonic acid group did not disturb the crystalline arrangement of graphene [18].

Fourier transform infrared spectra of PVDF membranes in Figure 4 shows absorption bands at 3023, 1628, 1401, 1276, 1181, 975, 1070, 796, 762, 841, 573 and 613 cm^−1^. The existence of PVDF was recognized by a peak present at 1169 cm^−1^ for C–F stretching vibration as shown in Figure 4. The vibration peak observed at 1402.45 cm^−1^ was related to the bending vibration of the CH_2_ group and the vibration band present at 1071 cm^−1^ confirmed the presence of β crystalline segment of polyvinylidene fluoride. The vibrational bands present at 877 and 837 cm^−1^ were related to the rocking mode of vinylidene moiety of polyvinylidene fluoride. The bands that exist at 598 cm^−1^ were caused by bending vibration of the CF_2_ group [28].

The pristine polyvinylidene fluoride shows peaks present at 2θ = 18.4°, 38.7° and 20.8° which are related to planes of (020), (002) and (110) showing the indication of α and β segment correspondingly. In the XRD plot, it is observed that one strong peak is present at 2θ = 20° and rest three faint peaks are present at 2θ = 18.3° and 38.7° as shown in Figure 5. The peaks at 2θ = 20° and 2θ = 38.7° correspond to (110) and (211) planes of γ-polyvinylidene fluoride crystalline phase, and the peak exist at 2θ = 18.3° corresponding to (020) plane of α-polyvinylidene fluoride phase [29].

### 3.2. Morphology of Fabricated Membranes with Scanning Electron Microscopic Analysis

The scanning electron microscopic analysis determines the morphology and microstructures of the membrane. Figure 6 shows the SEM analysis of (M1) pristine PVDF, (M2) PVDF-1%PVP, (M3) SGO-0.1%, (M4) SGO-0.2%, (M5) SGO-0.3%, (M6) SGO-0.4 % and (M7) SGO-0.5% membranes. It is observed from figures that the thin microporous upper layer of membrane comprised of homogenous smooth structure and porous sublayer comprised of large voids. This arrangement could be due to higher mutual diffusivity of water and DMAc solvent. In the SGO doped membranes, cylindrical macrovoids stretched over and grown to be wider than pristine PVDF membrane. This could be due to a greater affinity for sulfonated graphene oxide with hydrophilic groups, which improves mass transfer among non-solvent and solvent through phase inversion method. The water permeability improves due to this process. This result shows that the pristine PVDF membrane has a smoother surface. The varying morphology of membranes could be explained on the basis of a phase separation event in immersion precipitation [30]. The SGO additives contributed towards widening the finger-like structure due to its polar nature. The hydrophilic nature of SGO affects its solvating capacity in the solvent, enhancing the PVDF solvency in casting solution, accelerates mass transfer of non-solvent (water) inside the polymer [5].

Thermo gravimetric investigation was performed to examine the thermal stability of nanocomposite membranes. Each membrane was subjected to heat ranging 25 °C to 700 °C at 10 °C /min. The TGA plot in Figure 7 illustrates mass degradation or weight loss between specific temperature ranges. The higher weight deprivation of SGO was ascribed to the greater bond-making ability of the –SO_3_H group with a water molecule. The weight release over 200 °C–300 °C was due to SGO because of the discharge of –SO_3_H groups from the surface of graphene oxide. The major weight loss at about 450 °C to 500 °C was attributed to thermal decomposition of polymeric backbone. The SGO membranes presented more enhanced thermal constancy than pure PVDF. The TGA curve of pristine PVDF membranes showed better performance than all sulfonated graphene oxide doped PVDF membranes. There is no early mass loss that takes place in PVDF [5].

The pore size and pore structure were controlled by the incorporation of sulfonated graphene oxide. In comparison with the pristine PVDF membrane, the surfaces of all SGO doped membrane showed a greater porosity because of mixing of fillers in the phase inversion procedure. Fascinatingly, the M6 membrane has greater porosity than M7 as shown in Figure 8. The reason is that, during the phase inversion method, an improved hydrophilic character of sulfonated graphene oxide enhanced solvent transfer from the polymeric material to water, which makes the possibility of formation of greater pore density. The M7 showed lower porosity than the M6 membrane due to the greater viscosity of the solution which forbade the pore formation during the phase inversion process [31]. At the SGO concentration 0.4 wt%, the porosity reached a maximum value of 93% and much increased in comparison to the pristine PVDF membrane. Alternatively, if the content of sulfonated graphene oxide was raised more, a denser formation in the sub layer with little cylindrical pores was formed. It is possible owing to the aggregation of nanomaterials and greater viscosity of polymeric casting solution by dispersion of excess SGO [5].

Mechanical strength of pristine and SGO doped PVDF membranes is measured to check the long-run constancy of membranes. Figure 8 shows that tensile strength of membranes increases as the concentration of SGO increases in the polymeric matrix. This result is owed to the strong attraction between sulfonated graphene oxide nanomaterials and molecular chains of PVDF which facilitates better interaction of nanoparticles in polymeric matrix. The tensile strength increases from M3-0.1% SGO (1.19) to M7-0.5% (1.40) by increasing the concentration of SGO nanoparticles from 0.1% to 0.5% as shown in Figure 8. This result shows that by increasing the SGO concentration, membranes become stiffer and stronger [32]. However, the tensile strength of the pure polymeric membrane is greater than the polymeric doped membrane. The tensile potency of pristine polyvinylidene fluoride membrane is 2.23 which further increases with the incorporation of 1% PVP polymer in the pure PVDF membrane [24].

Hydrophilicity and hydrophobicity of membranes can be determined through contact angle dimensions. Hydrophilicity of a membrane indicates the antifouling behavior of membranes. If a membrane has less contact angle (<90°) then it is more hydrophilic and shows great resistance against fouling. If a membrane has more contact angle (>90°), the membrane has hydrophobic nature, and it shows less resistance against fouling. Generally, the high hydrophilic character of the membrane surface is described by a low contact angle. The addition of hydrophilic nanomaterials generally reduces the water contact angle with membrane. From Figure 9, it could be concluded that water contact angle was declined from 69.19 to 51.81 °C with the addition of SGO content in PVDF matrix. It means the hydrophilic character of the membrane surface was improved which is ascribed by the existence of (–SO_3_H) groups on SGO additives which exhibits higher water uptake level and occurrence of abundant hydrophilic functional groups [33]. The lowest contact angle and thus higher hydrophilicity is shown by M7 among all the nanocomposite membranes due to the addition of hydrophilic SGO. 

The water flux of the pristine PVDF and PVDF/SGO nanocomposite membranes were calculated using a dead-end filtration assembly (with a 0.037 m effective membrane diameter and 1.074 × 10^−3^ m^2^ effective membrane area) at 25 °C. Pure water permeability/flux of fabricated PVDF-SGO hybrid membranes is shown in Figure 9. The highest permeability was obtained by M7 PVDF-SGO (0.5 wt %) membrane. This membrane possessed the lowest contact angle with the highest hydrophilicity. The lower contact angle means more permeability to water and thus results in higher flux due to the occurrence of more hydrophilic sulfonated graphene oxide. Thus, it can be concluded that water flux of prepared membrane was considerably enhanced through rising hydrophilic SGO additives concentration labeled as M3, M4, M5, M6 and M7, respectively. The highest water flux is shown by M7 membrane with (0.5%) concentration which is up to 308.7 Lm^−2^ h^−1^ /bar in comparison to the pure PVDF membrane with 112.3 L m^−2^ h^−1^ /bar pure water flux as shown in Figure 9. It is estimated from results that the water contact angle and water flux /permeability have inverse relation, i.e., by increasing the water contact angle, water flux decreases due to less hydrophilic nature of membrane.

Cross flow filtration tests were conducted for each membrane with bovine serum albumin solution. The results of bovine serum albumin solution rejections are shown in Figure 10. Protein molecules attached the membrane surface and blocked the pores as BSA is hydrophobic in nature. However, SGO membranes showed higher values of BSA rejections. The M6 (0.4% SGO) represents the highest value of BSA rejection than all other membranes which is ascribed to the existence of more concentration of SGO, more hydrophilicity. However, M7 (0.5% SGO) have more concentration of SGO than M6 (0.4% SGO). This trend can be elucidated by the truth that by increasing the concentration of SGO, the polymeric solution becomes more viscous, resulting in pore blockage due to higher concentration of nanoparticles which results in aggregation of nanoparticles [5]. The salt rejection for pristine PVDF and PVDF-PVP membranes is less than the sulfonated graphene oxide doped polymeric membranes due to the absence of SGO as represented in Figure 10. The M7 exhibits the highest rejection of salts as compared to other nanocomposite membranes. These results showed that the nanocomposite membrane can be utilized for salt rejection from sea water desalination [24].

The antibacterial activity of the membrane was calculated by zone inhibition method. Clear rings formed around the membranes showing that the bacterial growth was inhibited by the membranes against *E. coli* and *S. aureus*. Figure 11 and Figure 12 show that the nanocomposite membranes represent larger inhibition zones. Figure 11 and Figure 12 illustrated that with the increase in sulfonated graphene oxide concentration, the inhibition zone of bacterial growth increased [6,24]. This is due to the presence of reactive oxygen species like O^2−^, H_2_O_2_ and OH^−^ generated by sulfonated graphene oxide, which harms bacterial DNA and is key for its antibacterial activity [34,35]. Furthermore, the transcriptional regulatory mechanism behind the enhancement of anti-biofouling performance is also important from sustainability perspectives [36].

## 4. Conclusions

Biofouling is one of the major hurdles in the broader application of membrane technology for wastewater treatment. Therefore, modification of membrane with anti-biofouling material seems to be an effective treatment for the reduction in adverse effects of such phenomenon. Due to superior antibacterial activity of SGO nanoparticles, its uniform dispersion within PVDF polymeric matrix imparts excellent anti-biofouling properties to the final polymer membrane. In this regard, anti-bacterial SGO nanoparticles were synthesized and used for the preparation of PVDF-based nanocomposite anti-biofouling membrane by phase inversion method. These fabricated membranes showed high performance such as high water flux of about 308.7 Lm^−2^h^−1^/bar with high BSA rejection as well as high mechanical strength and more hydrophilic character is exhibited by these membranes. Additionally, these membranes have high anti-bacterial activity against E. coli. It is confirmed from obtained results that our approach can guarantee successful anti-biofouling and good separation performance of SGO-based PVDF membrane in comparison to pristine PVDF membrane.

## Figures and Tables

**Figure 1 membranes-11-00749-f001:**
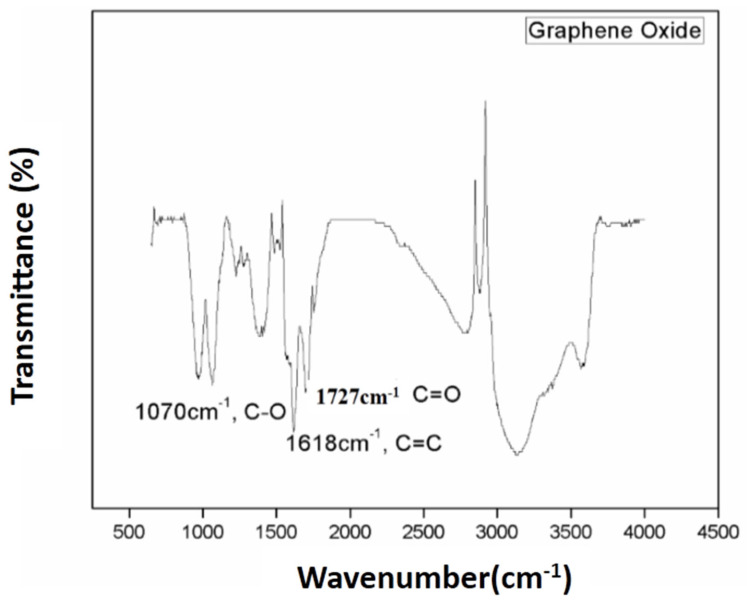
FTIR spectrum of graphene oxide.

**Figure 2 membranes-11-00749-f002:**
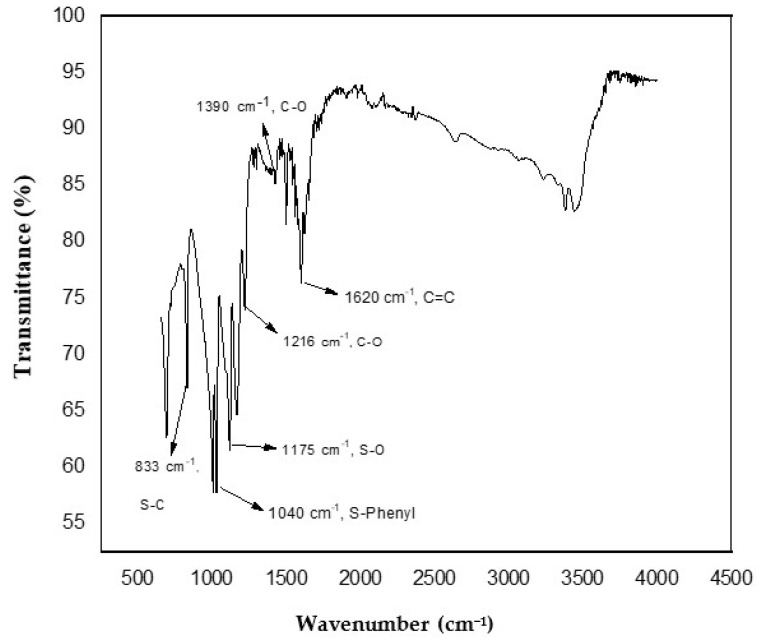
FTIR spectrum of sulfonated graphene oxide.

**Figure 3 membranes-11-00749-f003:**
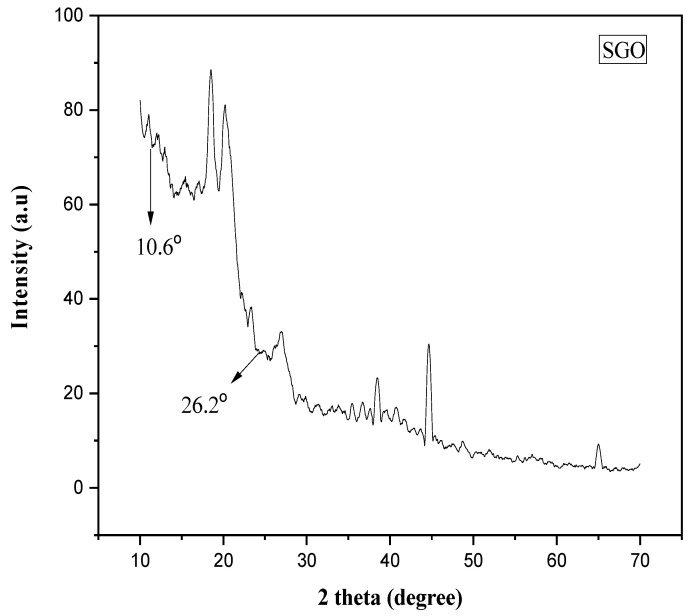
XRD spectrum of sulfonated graphene oxide.

**Figure 4 membranes-11-00749-f004:**
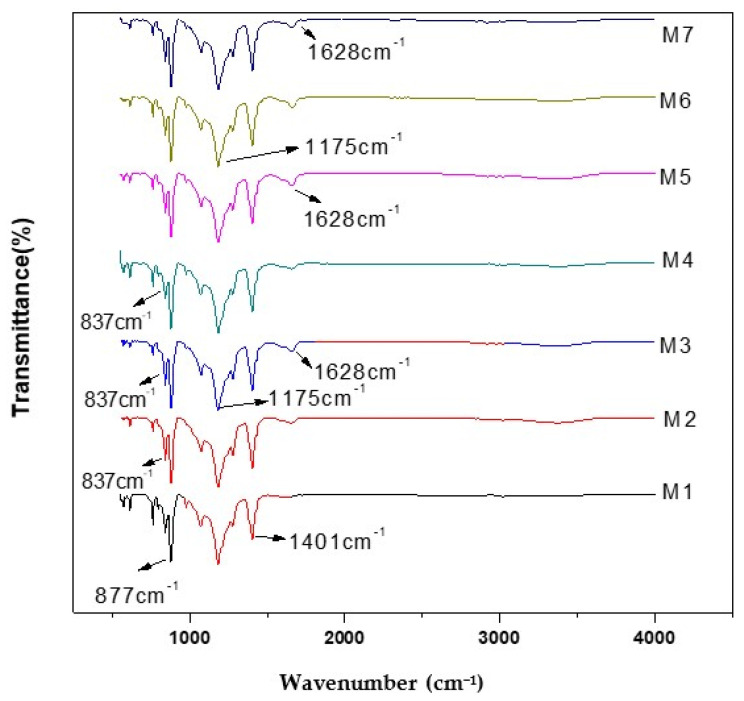
FTIR spectra of nanocomposite membranes.

**Figure 5 membranes-11-00749-f005:**
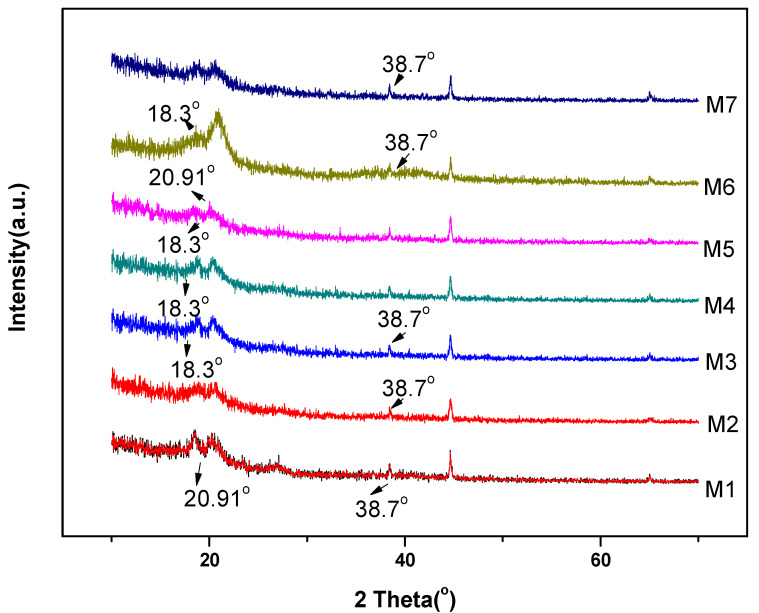
XRD spectra of nanocomposite membranes.

**Figure 6 membranes-11-00749-f006:**
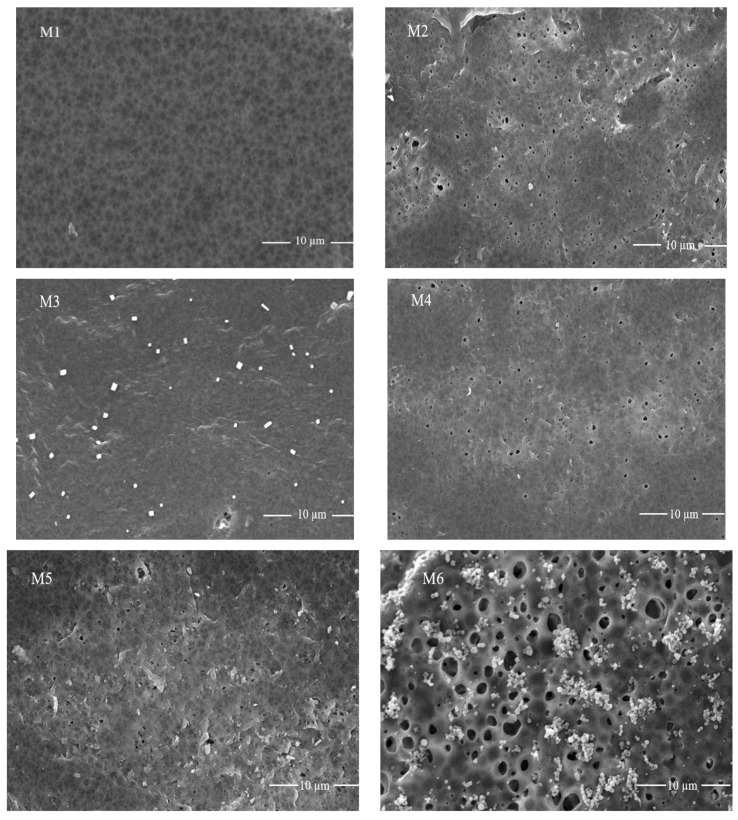
SEM analysis for the surface morphological studies of nanocomposite membranes: Top four membranes from the left named as M1 (PVDF only), M2 (PVDF-1% PVP), M3 (0.1% SGO), M4 (0.2% SGO) and bottom three as: M5 (0.3% SGO), M6 (0.4% SGO) and M7 (0.5% SGO).

**Figure 7 membranes-11-00749-f007:**
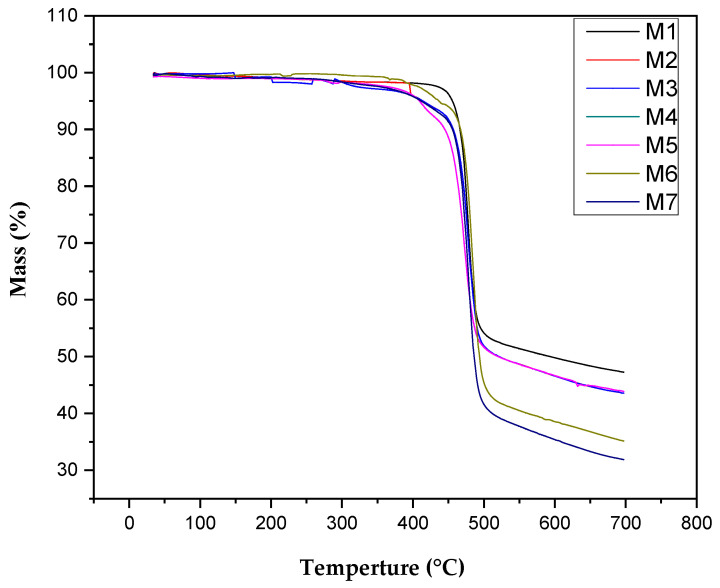
Thermogravimetric analysis of nanocomposite membranes.

**Figure 8 membranes-11-00749-f008:**
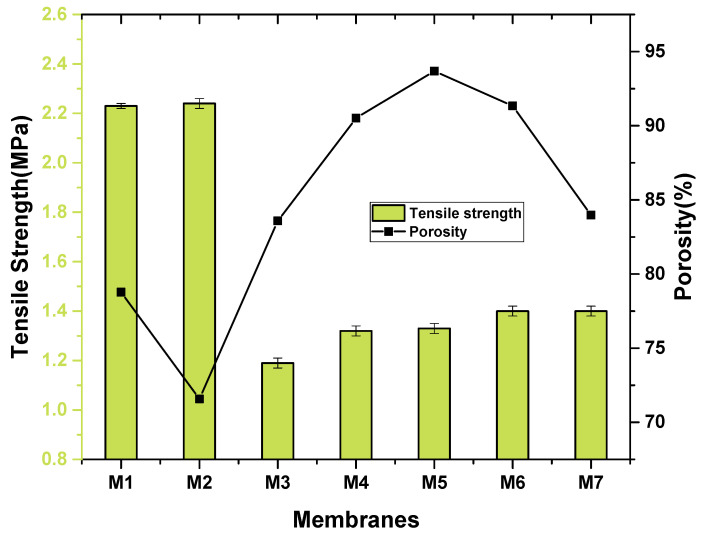
Porosity and tensile strength of nanocomposite membranes.

**Figure 9 membranes-11-00749-f009:**
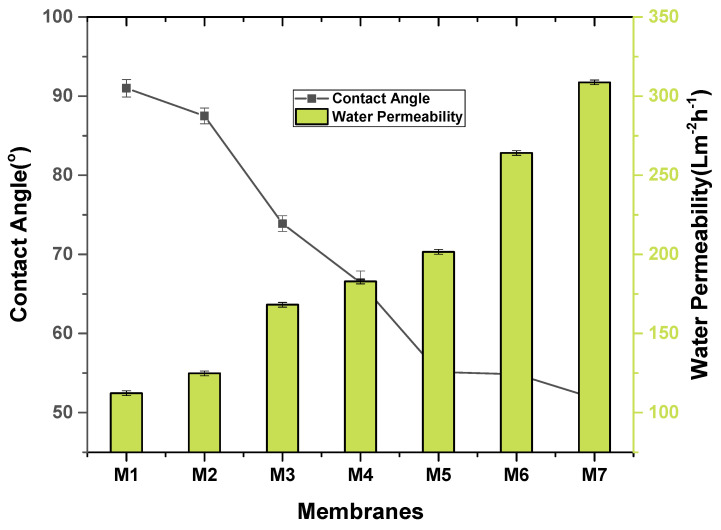
Contact angle and water permeability of nanocomposite membranes.

**Figure 10 membranes-11-00749-f010:**
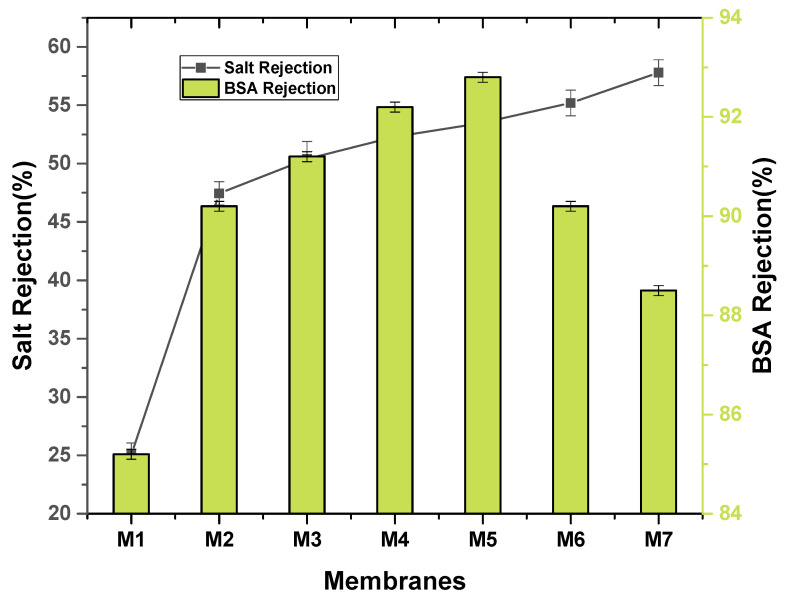
BSA and salt rejection of nanocomposite membranes.

**Figure 11 membranes-11-00749-f011:**
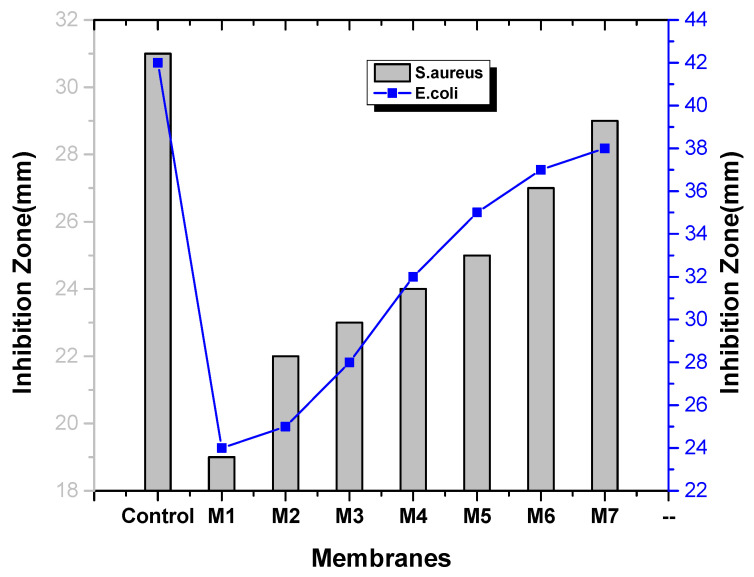
Anti-bacterial activity of nanocomposite membranes.

**Figure 12 membranes-11-00749-f012:**
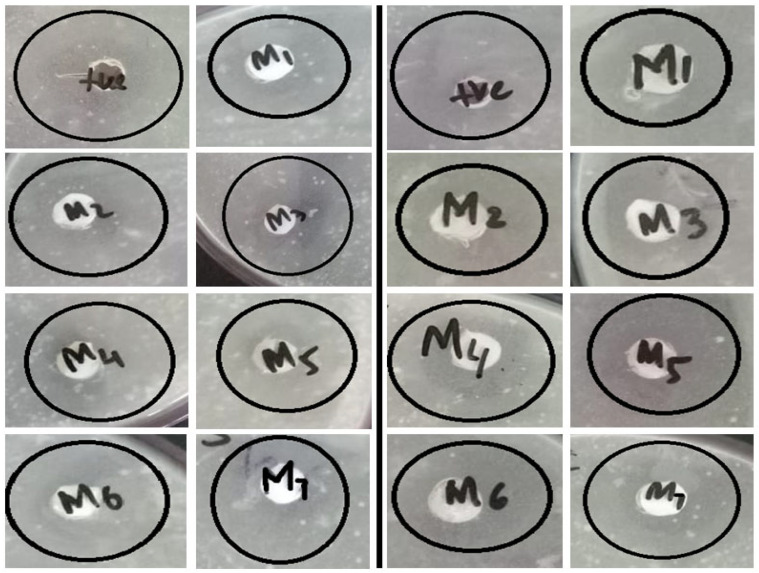
Nanocomposite membranes showing inhibition zones against *E. coli* (**left**) and *S. aureus* (**right**).

**Table 1 membranes-11-00749-t001:** Composition of different nanocomposite membranes.

Nanocomposite Membranes	SGO (wt %)	PVDF (wt %)	PVP (wt %)	DMAc (wt %)
M1	-	15	-	85
M2	-	15	1	84
M3	0.1	15	1	84.9
M4	0.2	15	1	84.8
M5	0.3	15	1	84.7
M6	0.4	15	1	84.6
M7	0.5	15	1	84.5

## Data Availability

Not applicable.

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
