# Peer review of "Fabrication and Characterization of Sulfonated Graphene Oxide (SGO) Doped PVDF Nanocomposite Membranes with Improved Anti-Biofouling Performance"

_membranes, 2021, doi:10.3390/membranes11100749_

Round 1

Reviewer 1 Report

The authors have studied the preparation and characterization of sulfonated graphene oxide (SGO) doped PVDF nanocomposite membranes in order to improve their anti-biofouling and anti-bacterial behavior. The SGO-PVDF membranes were fabricated through phase inversion method and their structure and morphology were characterized by Fourier transform infrared spectroscopy (FTIR), scanning electron microscopy (SEM), X-rays diffraction (XRD) and thermo-gravimetric analysis (TGA) techniques. The performance of SGO-PVDF membranes were evaluated via pure water flux and the anti-biofouling behavior was determined through Bovine Serum albumin (BSA) rejection.

The authors found that the SGO-PVDF membranes showed high antibacterial activity, more hydrophilicity and mechanical strength as compared to pristine PVDF membranes.

Some revisions are necessary:

1. At page 1, lines 27-28: "Membrane was fabricated through phase inversion method and its structure and morphology was characterized by...". Because, FTIR, XRD and TGA are structural analysis.

2. At page 5, line 177: Please insert the model of the UV-spectrophotometer.

3. At page 8, lines 225-230: Usually, the FTIR peaks are presented as integer values (without decimals).

4. At page 9, line 233: Please correct Figure 4 instead of Figure 1.

5. At page 9, lines 237 and 240: In the Figure 5 are not visible (marked) the peak(s) 2θ = 26.1°.

6. At page 10, lines 245-246: Please use the same labels as defined in Table 1 (instead of a, b, c, d...).

7. At page 10, lines 244-259: Please explain/ comment the influence of SGO concentration on pores shape and dimensions.

8. At page 11, Figure 6: The detail pictures of SEM images seems to be different than selected areas. The scale bars of detail pictures are not clear.

9. At page 12, line 261: The Figure 6-M7 is horizontally deformed (stretched). Please fix it.

10. At page 12, lines 262-264: Please use the same labels as defined in Table 1 (instead of i, ii, iii, iv...).

11. At page 12, lines 271-275: Please explain/ comment the influence of SGO concentration on final residue of TGA curves and why pristine PVDF membranes showed better thermal performance than all sulfonated graphene oxide doped PVDF membranes. Also, please explain why the final residue vary so much (almost between 30% and 50% wt.) with SGO concentration (which vary so little – 0.1 to 0.5% wt.)? It would be useful to insert a table with the mass losses.

12. At page 13, line 293: "The Table 2 shows that..." Unfortunately, the Table 2 is missing from the manuscript.

13. At page 13, line 304 (Figure 8): The authors must insert a secondary vertical axis (Tensile Strength [MPa]).

14. At page 14, lines 318-320: The statement "The lowest contact angle and thus higher hydrophilicity is shown by M7 among all the nanocomposite membranes due to addition of hydrophilic SGO." is in disagreement with Figure 9, where the water contact angle (WCA) of M7 is higher that M6.

15. At page 14, lines 324-325: The statement "The highest permeability was obtained by M7 PVDF-SGO (0.5wt %) membrane." is in disagreement with Figure 9, where the permeability of M7 is lower that M6.

16. At page 14, line 330: Please use the same labels as defined in Table 1 (instead of SGO-0.1%, SGO-0.2%, SGO-0.3%, SGO-0.4% and SGO-0.5%).

17. At page 14, line 336 (Figure 9): Please use the same labels as defined in Table 1 (instead of M0, M 1%PVP, M 0.1,...). Why the WCA of PVDF is so small (69.19°)? It is well known that the PVDF is highly hydrophobic material, with WCA > 90° for sure (see the definition of hydrophobic surfaces), almost of 120°. Please repeat this measurement.

18. At page 15, line 339-340: Please correct Figure 10 instead of Figure 3.

19. At page 15, line 342-343: The statement "The M5 (0.3% SGO) represents the highest value of BSA rejection than all other membranes" is in disagreement with Figure 10, where the value of BSA rejection of M5 is lower that M6.

20. At page 15, lines 350-351: "...due to the absence of SGO"

21. At page 15, line 354 (Figure 10): Please use the same labels as defined in Table 1 (instead of M 0, M 1%PVP, M 0.1,...).

22. At page 16, line 364 (Figure 11): Please use the same labels as defined in Table 1 (instead of M 0, M 1%PVP, M 0.1,...).

23. At page 17, line 367: The legend of Figure 12 must be more clear (e.g. which samples are seeded with S. aureus and which with E. coli; left or right?).

24. At page 17, lines 369-380 (Conclusion): The Conclusion section should be more detailed.

Author Response

Response is attached 

Reviewer 2 Report

Comment:

The effect of sulfonated graphene oxide (SGO) on the biofouling activity and other physical properties of PVDF membrane was investigated in this study. The authors demonstrated that the functionalization of PVDF membrane with SGO did improve some of its properties, such as surface hydrophilicity, porosity, water permeability and anti-biofouling activity. Although this is a well written manuscript, there are some important issues still need to be clarified, explained and discussed at this stage. Therefore, this paper may be published in this journal after addressing the following issues:

Specific comments:

1) The rationale of using PVP in the preparation of SGO-doped PVDF nanocomposite membranes should be described in the manuscript? The role of PVP in the microstructure and pore structure of the PVDF membranes (M1 and M2) should be discussed?

2) The Figure 1 at p.9 should be Figure 4.

3) (Figures 1 and 4) The wavenumbers assigned to functional groups in these FTIR spectra and in the main text should be consistent.

4) In Figure 4 the authors have measured the FTIR spectrum of PVDF samples doped with various amounts of sulfonated graphene oxide (M1-M7). The authors should explain and discuss the differences they discovered in these FTIR spectra.

5) Please explain why the peaks at 18.3˚ and 20.91˚ were disappeared in the Raman spectrum of M4.

6) The quality of the SEM images in Figure 6 is unacceptable. The scale bar and details about the microstructures were rather fuzzing. The authors should provide high quality SEM images for reviewing. On the other hand, the image of the inset in each panel is dissimilar to that enclosed in the white square area of each SEM image with low magnification power. The authors should explain why this may be happened.

7) The authors claimed that the SGO membranes exhibited “enhanced thermally constancy” than pure PVDF. The authors should explain based on what arguments to determine that the SGO membranes presented “enhanced thermally constancy”. How can PVDF membrane be benefited from having “enhanced thermally constancy”? The thermal

8) (lines 282-286) There is no difference in meaning between the phrases “the M6 membrane has greater porosity than M7” and “The M7 showed lower porosity than M6 membrane”. Then, how come the authors gave two distinct reasons to explain the same event? Is there any evidence to show that the PVDF with more SGO exhibit higher viscosity? Are the increasing of viscosity increase and the increasing of solvent transfer capability from polymer to water upon the increase of SGO content correlated or independent events? The authors should verify and discuss this issue in the manuscript.

9) The PDVF membrane samples with different SGO content were designated as M1 to M7. This marking system should be adopted throughout the manuscript. Hence, the caption of x-axis in Figures 9 and 10 and the related content in the manuscript should be revised.

10) Figure 9 shows that the M6 instead of M7 exhibits lowest contact angel and highest water permeability. The description in line 319 should be revised. Although M7 received the highest amount of SGO during the preparation, its surface hydrophilicity was lower than that of M6 as evidenced by the higher contact angel than that of M6. The authors should comment on this.

Author Response

The comments are attached 

Reviewer 3 Report

Dear Editor,

I have reviewed the article entitled “Fabrication and Characterization of Sulfonated Graphene Oxide (SGO) doped PVDF Nanocomposite Membranes with Improved Anti-biofouling Performance" by Muhammad Zahid, Tayyaba Khalid, Talha Javed, Saba Akram, Anum Rashid, Zulfiqar Ahmad Rehan, Rubab Shabbir, Muhammad Shoaib Asad, Mohamed M. Hassan, Mahmoud M. Hessien (Manuscript ID membranes-1374875).

Here my comments:

In general, the article needs a major revision before resubmission: although the English is Ok, the manuscript is very technical, describing several phenomena, without providing sufficient support and explanations.

The introduction part is too short, without showing the relevancy of the research and explanation of the materials selection. There are many quotations however their relevancy to the text is limited.

The graphene platelets may act as a physical barrier. Please refer to the selection of graphene.

It is recommended to add more relevant manuscripts describing similar phenomena using conductive polymers, such as:

  • Chajanovsky, and R. Y. Suckeveriene, Preparation of Hybrid Polyaniline/Nanoparticle Membranes for Water Treatment Using an Inverse Emulsion Polymerization Technique Under Sonication, Process MDPI 8(11), 1503 (2020), https://doi.org/10.3390/pr8111503

Figure 1 & 2 (FTIR spectra) should be together in the same curve.

Page 9 – there are two “Figure 1”. Please change the numbering. Also, the FTIR curves in page 9 are similar, thus this graph can be excluded or published as supplementary data.

Page 12, Figure 7, Please add the DTG curve for the TGA measurement. Also, please refer to the residual mass at the end of the experiment.

There is still work to be done regarding the context. The manuscript is somewhat shallow. The discussion mainly describes the results and lacks explanations. It is recommended that the authors will review the text before re-submission.

Major revision is needed before resubmitting the manuscript.

Author Response

The response is attached 

Reviewer 4 Report

Membrane technologies play essential role in water purification techniques, so the work addressing a new approach of doping of polymeric material (PVDF) by sulfonated graphene oxide in terms of achieving the high anti-biofouling and anti-bacterial behavior is quite in need. There are several interesting findings demonstrating that, along with the high anti-bacterial activity, the SGO dopped PVDF exhibits higher water flux, more hydrophilicity, as well as higher mechanical strength. This combination of properties makes the modified PVDF a promising tool in water purification applications.
The strength of the work: State of the art sample preparation techniques in combination of an appropriate complimentary analytical tools enabling achieving quite interesting and unambiguous results.
The weakness: The work would gain if the results were added by characterization of the samples in terms of local atomic/molecular content and at the SGO-PVDF interface by means of e.g. XPS. 
In general, the work is scientifically sound, is of quite high level, logically and clearly performed and presented, reference list is appropriate and up-to-dated, figures are informative and clear. In my view, the manuscript is suitable for publication in Membranes in its present form.

Author Response

The response is attached 

Round 2

Reviewer 1 Report

The authors improved their manuscript and performed most of the suggested corrections. However, there are still two (missed) modifications to do, before to be accepted for publication in Membranes journal:

1. At page 9, line 241: The figure’s number must be changed from 1 in 4.

2. At page 17, line 383 (Figure 11): The Ox labels of the Figure 11 must be the same used along the manuscript and defined in Table 1.

Author Response

The authors improved their manuscript and performed most of the suggested corrections. However, there are still two (missed) modifications to do, before to be accepted for publication in Membranes journal:

  1. At page 9, line 241: The figure’s number must be changed from 1 in 4.

Esteemed reviewer, the changes have been incorporated accordingly.

  1. At page 17, line 383 (Figure 11): The Oxlabels of the Figure 11 must be the same used along the manuscript and defined in Table 1.

Thanks for the comments. The changes have been incorporated accordingly.

Reviewer 2 Report

1) It doesn't matter whether PVP causes any change in chemical and physical properties of M1-M7. There must be a reason for the authors to include PVP in the fabrication of the PVDF membranes. The authors should discuss this issue in the manuscript.

2) The title of the figure legend (p.9) in Figure 4 was not changed.

3) In Figure 4 the authors have measured the FTIR spectrum of PVDF samples doped with various amounts of sulfonated graphene oxide (M1-M7). There is no sulfonated graphene oxide in M1 and M2. Based on Figure 2, two peaks at 1175 cm-1 and 1126 cm-1 (S-O) and peak at 1040 cm-1 (S-phenyl) could be found in the FTIR spectrum of sulfonated graphene oxide. Why are there no differences in FTIR spectra were found between spectra of M1/M2 and M3-M7? The authors should comment on this issue. 

Author Response

  1. It doesn't matter whether PVP causes any change in chemical and physical properties of M1-M7. There must be a reason for the authors to include PVP in the fabrication of the PVDF membranes. The authors should discuss this issue in the manuscript.

Response: Thanks for comments. PVP act as pore forming agent during phase inversion method. This is also discussed in manuscript and highlighted according to your suggestion.

  1. The title of the figure legend (p.9) in Figure 4 was not changed.

Response: Esteemed reviewer, the title is changed accordingly.

  1. In Figure 4 the authors have measured the FTIR spectrum of PVDF samples doped with various amounts of sulfonated graphene oxide (M1-M7). There is no sulfonated graphene oxide in M1 and M2. Based on Figure 2, two peaks at 1175 cm-1 and 1126 cm-1 (S-O) and peak at 1040 cm-1 (S-phenyl) could be found in the FTIR spectrum of sulfonated graphene oxide. Why are there no differences in FTIR spectra were found between spectra of M1/M2 and M3-M7? The authors should comment on this issue. 

Response: Thanks for the comments. PVP is used in less amount that’s why M1 and M2 are same, whereas due to same composition, M3-M7 are same because concentration do not change peaks.
